# Phase shift in skyrmion crystals

Satoru Hayami ⬤ [1✉], Tsuyoshi Okubo ⬤ [2] & Yukitoshi Motome ⬤ [1]

The magnetic skyrmion crystal is a periodic array of a swirling topological spin texture. Since it is regarded as an interference pattern by multiple helical spin density waves, the texture changes with the relative phase shifts among the constituent waves. Although such a phase degree of freedom is relevant to not only magnetism but also transport properties, its effect has not been elucidated thus far. We here theoretically show that a phase shift in the skyrmion crystals leads to a tetra-axial vortex crystal and a meron-antimeron crystal, both of which show a staggered pattern of the scalar spin chirality and give rise to nonreciprocal transport phenomena without the spin-orbit coupling. We demonstrate that such a phase shift can be driven by exchange interactions between the localized spins, thermal fluctuations, and long-range chirality interactions in spin-charge coupled systems. Our results provide a further diversity of topological spin textures and open a new field of emergent electromagnetism by the phase shift engineering.

[1] Department of Applied Physics, The University of Tokyo, Tokyo, Japan. [2] Institute for Physics of Intelligence, The University of Tokyo, Tokyo, Japan.
✉email: hayami@ap.t.u-tokyo.ac.jp

The skyrmion is a topological configuration of a continuous field. Although it was originally proposed to explain hadrons in the particle theory[1,2], it has turned out to be realized in various forms in condensed matter physics[3]. One possible realization was discovered in magnets, in the form of skyrmion-like magnetic textures[4–8]. The magnetic skyrmions often exist in their stable form, the so-called skyrmion crystal (SkX), which is a periodic array of the particle-like magnetic skyrmions. Importantly, such a SkX is approximately expressed as a superposition of multiple helical spin density waves, and hence, it can be regarded as an interference pattern by the multiple helices. It has attracted enormous attention since the swirling magnetic texture generates an emergent magnetic field through the Berry phase mechanism and results in peculiar transport phenomena, such as the topological Hall effect[3,9,10].

Similar to an isolated skyrmion, the SkX is characterized by three quantities: skyrmion number, vorticity, and helicity[3]. However, as the SkX is regarded as an interference pattern, it has another degree of freedom, which has been overlooked in the previous researches, the phases of the constituent waves. This is exemplified for three scalar waves in Fig. 1a, b, where a phase shift in one of the three waves leads to a different interference pattern with different symmetry. Such a phase degree of freedom exists in all the interference phenomena, except for linearly independent waves in continuous space for which a phase shift is equivalent to a spatial translation. The SkX appears not in a continuous field but for spins on a discrete lattice, which leads to a further variety of the interference patterns by the discretization, even for the linearly independent waves. A shift of the relative phases changes not only magnetic textures but also emergent magnetic fields, and hence, transport properties, but such an interesting possibility has not been elucidated thus far.

In this study, we theoretically unveil the effect of phase shifts in the SkX and propose how to control the phase degree of freedom. Considering an itinerant electron model on a triangular lattice, we show that the SkX turns into a tetra-axial vortex crystal (TVX) or a meron-antimeron crystal (MAX) by a phase shift of $\pi/2$. The phase-shifted states have distinct properties from the SkX: The SkX exhibits a net scalar chirality leading to the topological Hall effect, while the TVX and MAX exhibit a staggered one that does not lead to the topological Hall effect, but induces nonreciprocal transport phenomena that do not require the spin-orbit coupling. We find that such a phase shift can be caused by several different mechanisms, such as exchange interactions between the localized spins, thermal fluctuations, and long-range chirality interactions. Our results open another route to a further variety of magnetic textures which have been overlooked in skyrmion-hosting materials.

## Results

Let us start by classifying noncoplanar spin textures according to the type of constituent waves and the relative phases. First, we consider a superposition of spiral spin textures represented by $\mathbf{S}_i^{\text{spiral}} = \sum_{\nu=1}^{3} \left( \sin \mathcal{Q}_\nu \cos \phi_\nu, \sin \mathcal{Q}_\nu \sin \phi_\nu, -\cos \mathcal{Q}_\nu \right)$, where $\mathcal{Q}_\nu = \mathbf{Q}_\nu \cdot \mathbf{r}_i + \theta_\nu$ and $\phi_\nu = \frac{2}{3}\pi(\nu - 1)$; $\mathbf{Q}_\nu$ and $\theta_\nu$ are the wave vector and the phase of the $\nu$th spiral, respectively, and $\mathbf{r}_i$ is the position vector for site $i$. In the following analyses, as an archetype, we consider a two-dimensional triangular lattice system with threefold rotationally symmetric wave vectors with spiral pitch $Q$: $\mathbf{Q}_1 = (Q, 0)$, $\mathbf{Q}_2 = (-Q/2, \sqrt{3}Q/2)$, and $\mathbf{Q}_3 = (-Q/2, -\sqrt{3}Q/2)$ satisfying $\sum_\nu \mathbf{Q}_\nu = 0$. The spin texture $\mathbf{S}_i^{\text{spiral}}$ is modulated by shifting the phases $\theta_\nu$. We demonstrate the situation for $\theta_1 = \theta_2 = \theta_3$ by changing the

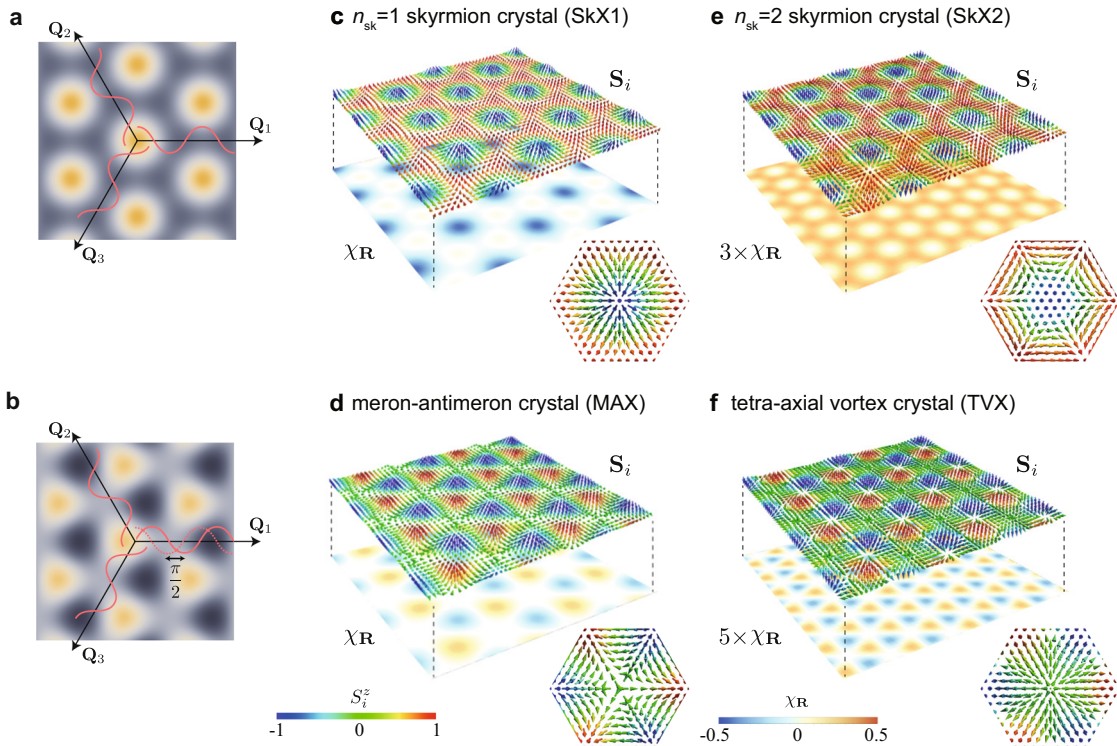

**Fig. 1 Phase shift and interference patterns. a, b** Superpositions of three density waves with the wave vectors $\mathbf{Q}_1$, $\mathbf{Q}_2$, and $\mathbf{Q}_3$. **b** is the figure generated from **a** with the phase shift of $\pi/2$ in $\mathbf{Q}_1$, which breaks sixfold rotational symmetry in **a**. **c, d** superpositions of three spirals waves: the $n_{\text{sk}} = 1$ skyrmion crystal (SkX1) (**c**) and the meron-antimeron crystal (MAX) (**d**). **e, f** Superpositions of three sinusoidal waves: the $n_{\text{sk}} = 2$ skyrmion crystal (SkX2) (**e**) and the tetra-axial vortex crystal (TVX) (**f**). **d, f** are generated from (**c**) and (**e**), respectively, with the phase shift of $\Theta = \pi/2$, and both of them break sixfold rotational symmetry similar to **b**. In **c–f**, the upper and lower planes show the spin $\mathbf{S}_i = (S_i^x, S_i^y, S_i^z)$ [the color scale indicates $S_i^z$, and the arrows indicate $(S_i^x, S_i^y)$] and the scalar chirality $\chi_{\mathbf{R}}$, respectively. The spin textures in the magnetic unit cell are shown in the insets of **c–f**.

total phase $\Theta = \sum_\nu \theta_\nu$. Figure 1c, d displays the spin textures with $\Theta = 0$ and $\pi/2$. In each figure, the upper and lower planes show the textures of spin and scalar spin chirality, respectively; the latter is defined by $\chi_{\mathbf{R}} = \mathbf{S}_i \cdot (\mathbf{S}_j \times \mathbf{S}_k)$, where $\mathbf{R}$ represents the position vector at the center of a triangle with sites $i, j, k$ in the counterclockwise order. The case with $\Theta = 0$ in Fig. 1c is a periodic array of skyrmions with the skyrmion number of one ($n_{sk} = 1$), which we call the SkX1. It retains sixfold rotational symmetry in both spin and chirality, and has a nonzero net chirality leading to the topological Hall effect. Meanwhile, the case with $\Theta = \pi/2$ in Fig. 1d is a staggered arrangement of merons and antimerons (half skyrmions and antiskyrmions[11]), which we call the MAX. In this state, the rotational symmetry is reduced to threefold. Moreover, the meron and anti-meron carry the skyrmion number of $+1/2$ and $-1/2$, respectively, and hence, the total skyrmion number is zero in the MAX; accordingly, the net value of $\chi_{\mathbf{R}}$, $\chi^{total} = \frac{1}{N}\sum_{\mathbf{R}}\chi_{\mathbf{R}}$ where $N$ is the number of lattice sites, also vanishes and the MAX does not show the topological Hall effect. We note that the MAX was proposed as a candidate for the unidentified magnetic state next to the SkX1 found in a triangular-lattice magnet $Gd_2PdSi_3$[11].

Next, we consider a superposition of sinusoidal waves[12–14], which is represented by $\mathbf{S}_i^{sin} = (\cos\mathcal{Q}_1, \cos\mathcal{Q}_2, \cos\mathcal{Q}_3)$. Similar to $\mathbf{S}_i^{spiral}$, different $\Theta$ gives different spin and chirality textures, as shown in Fig. 1e, f (the spin frame is rotated for better visibility). The spin texture with $\Theta = 0$ in Fig. 1e is the other SkX called the SkX2, in which each skyrmion has the skyrmion number of two ($n_{sk} = 2$). In this state, while the spin texture has threefold rotational symmetry, the chirality $\chi_{\mathbf{R}}$ is sixfold and the net value $\chi^{total}$ is nonzero, similar to the SkX1 in Fig. 1c. The phase shift by $\pi/2$ lowers the symmetry from sixfold to threefold, as shown in Fig. 1f; $\chi_{\mathbf{R}}$ has a staggered configuration with no net scalar chirality, similar to the MAX in Fig. 1d. In this state, the spin texture is given by a periodic array of four types of vortices; the vortex axes, which are defined by the vorticity for $xy$, $yz$, and $zx$ components of spins point to four corners of the tetrahedron (see Supplementary Information). Hence, we call the $\Theta = \pi/2$ state the TVX.

Thus, in both cases, the phase shift changes not only the spin texture but also the symmetry and topology. In particular, the net value of the scalar chirality $\chi^{total}$ is sensitively dependent on $\Theta$; the two types of SkXs at $\Theta = 0$ have nonzero values and cause the topological Hall effect, while the MAX and TVX at $\Theta = \pi/2$ have no net value and do not show the topological Hall effect. Interestingly, however, the breaking of sixfold rotational symmetry in $\chi_{\mathbf{R}}$ in the MAX and TVX leads to Fermi surface deformations as discussed below, which can induce direction-dependent non-reciprocal transport phenomena without the spin-orbit coupling.

The optimal values of the phases $\theta_\nu$ will be determined by multiple factors, such as lattice geometry and interactions between the spins. In the previous studies, the SkXs with $\Theta = 0$ are stabilized, e.g., by the Dzyaloshinskii−Moriya (DM)[6,15], four-spin[16–19], frustrated[20–22], and spin-charge interactions[14,23–25] on various lattices. The key question addressed here is what is the relevant parameter to cause a phase shift that leads to switching of magnetic, topological, and transport properties. In the following, we unveil three different mechanisms for such a phase shift, by taking an archetypal model for itinerant magnets hosting SkXs, the Kondo lattice model on a triangular lattice where both the SkX1 and SkX2 appear in the ground state (see "Methods").

We first demonstrate that a phase shift can be caused by introducing the exchange interactions between the localized spins described by $\mathcal{H}^{loc} = \sum_{ij} J_{ij}\mathbf{S}_i \cdot \mathbf{S}_j$ to the original Kondo lattice Hamiltonian $\mathcal{H}$. Considering the first-, second-, and third-neighbor interactions, $J_1$, $J_2$, and $J_3$ for $J_{ij}$, respectively, we perform a variational calculation to determine the ground-state phase diagram of

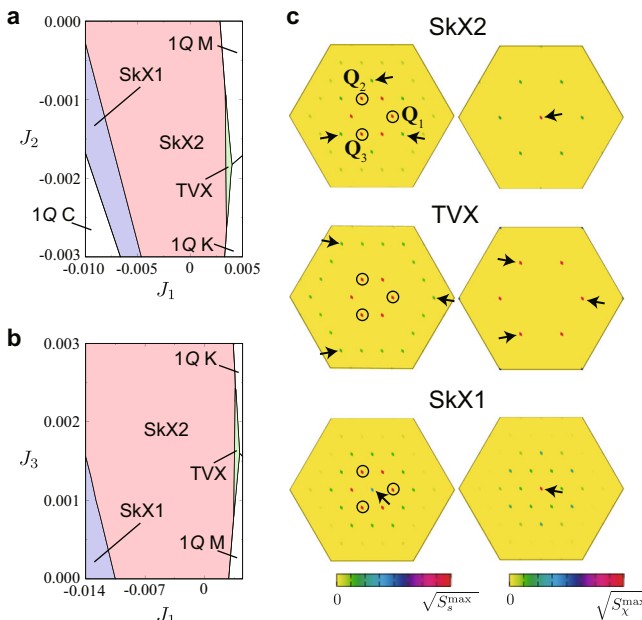

**Fig. 2 Variational ground states of the Kondo lattice model with the exchange interactions between the localized spins up to third neighbors, $J_1$, $J_2$, and $J_3$.** Variational phase diagrams on the $J_1−J_2$ (**a**) and $J_1−J_3$ (**b**) planes. **c** Spin (left) and chirality (right) structure factors for the SkX2 (top), TVX (middle), and SkX1 (bottom). The circles represent the peak positions at $\mathbf{Q}_1$, $\mathbf{Q}_2$, and $\mathbf{Q}_3$, and the arrows indicate the subdominant (dominant) peak positions in the spin (chirality).

the Hamiltonian $\mathcal{H} + \mathcal{H}^{loc}$ at zero field (see "Methods"). Figure 2a, b shows the results on the $J_1−J_2$ and $J_1−J_3$ planes. While the SkX2 is stable in the wide range of parameters, we find two topological phase transitions: One is to the TVX while increasing $J_1$ and decreasing $J_2$ (increasing $J_3$) and the other is to the SkX1 while decreasing $J_1$ and decreasing $J_2$ (increasing $J_3$), as shown in Fig. 2a (Fig. 2b). The former transition from the SkX2 to the TVX is accompanied by the phase shift with $\Theta = \pi/2$.

The phase transitions among these three states are understood from the higher harmonics in the spin structure. We show the spin structure factors in momentum ($\mathbf{q}$) space for the SkX2, TVX, and SkX1 in the left panels of Fig. 2c (see "Methods"). Although the dominant peaks appear at $\mathbf{Q}_1$, $\mathbf{Q}_2$, and $\mathbf{Q}_3$ commonly in the three phases, subdominant peaks are found at different $\mathbf{q}$ among them: $\mathbf{q} = \mathbf{Q}_\nu - \mathbf{Q}_{\nu'}$ ($\nu \neq \nu'$) in the SkX2, $\mathbf{q} = 3\mathbf{Q}_\nu$ in the TVX, and $\mathbf{q} = 0$ in the SkX1. Thus, considering that the Fourier transform of $\mathcal{H}^{loc}$ is written as $\sum_{\mathbf{q}} J_{\mathbf{q}} \mathbf{S}_{\mathbf{q}} \cdot \mathbf{S}_{-\mathbf{q}}$, the (anti)ferromagnetic interactions giving $J_0 < 0$ ($J_{3\mathbf{Q}_\nu} < 0$) tend to prefer the SkX1 (TVX).

It is noteworthy that the different superpositions of the constituent waves also give rise to the different peak positions in the $\mathbf{q}$-resolved scalar chirality shown in the right panels of Fig. 2c (see "Methods"). As mentioned above, both the SkX2 and SkX1 have the dominant peak at $\mathbf{q} = 0$ reflecting nonzero $\chi^{total}$, while the TVX has the dominant peaks at $2\mathbf{Q}_\nu$ with equal intensities and no weight at $\mathbf{q} = 0$.

The second mechanism to cause the phase shift is thermal fluctuations. We study the finite-temperature behavior of the Kondo lattice Hamiltonian $\mathcal{H}$ by performing the Langevin dynamics simulations with the kernel polynomial method (see "Methods")[26–28]. Figure 3a shows the results at zero magnetic field $H = 0$ where the ground state is the SkX2[14]. We find two phase transitions at $T_1 \simeq 0.0055$ and $T_2 \simeq 0.009$. The transition at $T_1$ is characterized by

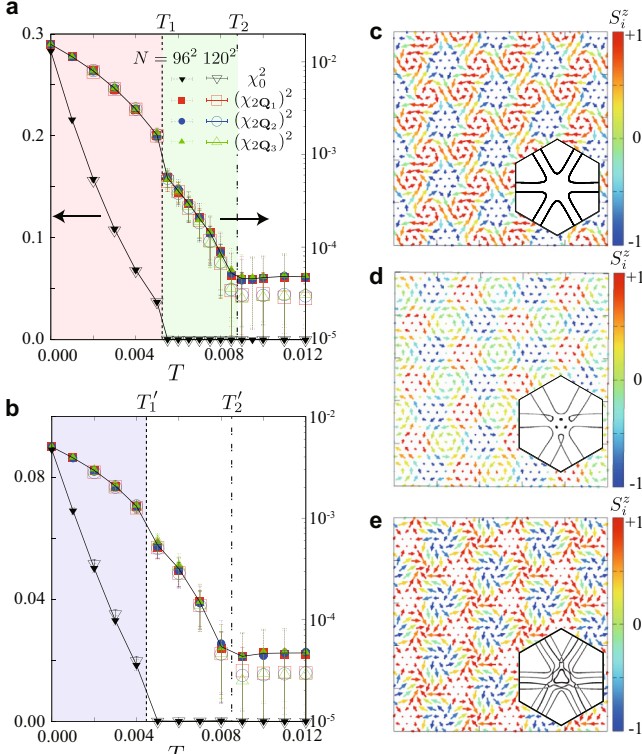

**Fig. 3 Finite-temperature behaviors of the Kondo lattice model.**
**a,b** Temperature dependences of the scalar chirality, $\chi_0^2$ and $\chi_{2\mathbf{Q}_\nu}^2$ at zero magnetic field $H = 0$ (**a**) and $H = 0.004$ (**b**). The dashed (dash-dotted) line represents the transition temperature $T_1$ ($T_2$) in **a** and $T_1'$ ($T_2'$) in **b**. **c–e** Real-space spin configurations in the SkX2 at $T = 0$ and $H = 0$ (**c**), the TVX at $T = 0.006$ and $H = 0$ (**d**), and the SkX1 at $T = 0$ and $H = 0.004$ (**e**). The color scale indicates $S_i^z$, while the arrows indicate $(S_i^x, S_i^y)$. The insets display the Fermi surfaces in each state; the sixfold rotational symmetry is broken in all the three states. See the main text.

the onset of $\chi_0$, suggesting that the SkX2 remains stable up to $T_1$. Note that the true magnetic long-range order is limited to zero temperature in the present two-dimensional system due to the Mermin−Wagner theorem[29]; the state for $0 < T < T_1$ is a chiral spin liquid having the SkX2 spin texture with a finite correlation length (but much longer than the system size). The real-space spin configuration at $T = 0$ is shown in Fig. 3c, which corresponds to that in Fig. 1e. Meanwhile, the transition at $T_2$ appears to be signaled by the onset of the higher harmonics $\chi_{2\mathbf{Q}_\nu}$ as plotted in Fig. 3a. A snapshot of the real-space spin configuration at $T = 0.006$ is shown in Fig. 3d, which well reproduces the spin texture for the TVX in Fig. 1f. From these results, we conclude that the low- and intermediate-temperature phases are chiral spin liquids with the SkX2 and TVX spin textures, respectively, and the transition at $T_1$ is associated with the phase shift of $\pi/2$ between them.

The appearance of the TVX at finite temperature is explained by an effective chirality interaction as follows. At the mean-field level, the entropic contributions are in general given in the form of $n$ th-order magnetic interactions as $T \sum_{\mathbf{q}_1 \cdots \mathbf{q}_n} (\mathbf{S}_{\mathbf{q}_1} \cdot \mathbf{S}_{\mathbf{q}_2}) \cdots (\mathbf{S}_{\mathbf{q}_{n-1}} \cdot \mathbf{S}_{\mathbf{q}_n}) \delta(\mathbf{q}_1 + \cdots + \mathbf{q}_n)$[30,31]. Among them, the lowest-order contribution to the phase shift appears in the sixth order. By considering $\mathbf{S}_i^{\sin}$, the dominant entropic contribution is given as $T\mathrm{Re}[(\mathbf{S}_{\mathbf{Q}_1} \cdot \mathbf{S}_{\mathbf{Q}_1})(\mathbf{S}_{\mathbf{Q}_2} \cdot \mathbf{S}_{\mathbf{Q}_2})(\mathbf{S}_{\mathbf{Q}_3} \cdot \mathbf{S}_{\mathbf{Q}_3})] = T\mathrm{Re}[\{\mathbf{S}_{\mathbf{Q}_1} \cdot (\mathbf{S}_{\mathbf{Q}_2} \times \mathbf{S}_{\mathbf{Q}_3})\}^2] \propto T\cos 2\Theta$. Thus, the sixth-order entropic term, which has the form of chirality-chirality interactions, depends on $\Theta$ and tends to stabilize the TVX at finite temperature.

Next, let us discuss the case of SkX1. Figure 3b represents the results under a magnetic field, where the ground state is the SkX1. Similar to the zero-field case in Fig. 3a, two phase transitions occur at $T_1' \simeq 0.0045$ and $T_2' \simeq 0.0085$. The low-temperature state below $T_1'$ is the SkX1 (with quasi-long-range order in the $xy$ components), whose spin configuration is shown in Fig. 3e[14]. Recalling the phase shift of $\pi/2$ from the SkX2 to the TVX in the zero-field case, one may expect that the SkX1 changes into the MAX by raising temperature, but we find that the intermediate phase for $T_1' < T < T_2'$ is a different super-position of three sinusoidal waves which has $\chi^{\text{total}} = 0$. This is because the Zeeman energy gain in the MAX is not sufficient to overcome the obtained state. We note that the threefold rotational symmetry is broken in the intermediate phase; hence, we call this phase the anisotropic 3Q state (see Supplementary Information). This phase transition from the SkX1 to the anisotropic 3Q state is also accounted for by the sixth-order entropic contribution, similar to that from the SkX2 to the TVX at zero field, as will be shown below.

We display the Fermi surfaces in the SkX2, TVX, and SkX1 in the inset of Fig. 3c–e, respectively. The Fermi surface in the TVX in Fig. 3d is threefold rotationally symmetric, meaning the breaking of the sixfold rotational symmetry of the triangular lattice, as expected from the above discussion. This leads to a nonreciprocal transport in itinerant electrons. Notably, there appear threefold rotationally symmetric Fermi surfaces even in the SkX2 in Fig. 3c (very weakly broken) and SkX1 in Fig. 3e. This is not due to the shift of $\Theta$ but by phase-locking at $(\theta_1, \theta_2, \theta_3) = (\pi/3, -\pi/3, 0)$ (any permutation is allowed) so that the skyrmion cores avoid the lattice sites (the values of $\theta_\nu$ depend on Q; see Supplementary Information). Thus, the results indicate that the individual phase $\theta_\nu$, as well as the total phase $\Theta$, are relevant in the actual discrete lattice systems.

The third mechanism is higher-order spin interactions, inferred from the above entropic mechanism. In general, the kinetic motion of itinerant electrons induces effective spin interactions, which can be explicitly derived by perturbation expansion in terms of the spin-charge coupling in the Kondo lattice model. The lowest-order contribution is a bilinear interaction called the Ruderman−Kittel−Kasuya−Yosida interaction[32–34], and the next fourth-order biquadratic inter-action was shown to be relevant to stabilize the SkXs[35]. We here consider a higher-order six-spin contribution given by $\tilde{L}[\{\mathbf{S}_{\mathbf{Q}_1} \cdot (\mathbf{S}_{\mathbf{Q}_2} \times \mathbf{S}_{\mathbf{Q}_3})\}^2 + \text{H.c.}]$ (see "Methods" and Supplementary Information). It is worthy to note that this has a similar form to the above six-spin entropic term, and it is the lowest-order contribution whose energy depends on $\Theta$ under $\sum_\nu \mathbf{Q}_\nu = 0$ in the perturbation expansion. This chirality inter-action is different from those discussed in the previous studies that stabilize noncoplanar spin states but appear to be irrelevant to the phase shift[36,37].

To clarify the effect of the chirality interaction, we investigate the ground-state phase diagram of the effective spin Hamiltonian $\mathcal{H}^{\text{eff}}$ by variational calculations and the simulated annealing (see "Methods"). We find that the SkX2 gives the lowest energy for $0 \le L < 1$, while the TVX does for larger $L$, as shown in Fig. 4a. The optimal $\theta_\nu$ are obtained as $(\theta_1, \theta_2, \theta_3) = (\pi/3, -\pi/3, 0)$ for the for-mer and $(\pi/3, \pi/6, 0)$ for the latter (any permutation is allowed), both of which are consistent with those in the Kondo lattice model above. Thus, the chirality interaction prefers the $\Theta = \pi/2$ states to the $\Theta = 0$ ones, namely, it brings about the phase shift in SkXs.

Furthermore, we find that the value of $L$ necessary for the phase shift can be largely reduced by combining the first mechanism by the exchange interactions for the localized spins. Figure 4b shows the ground-state phase diagram of the Hamil-tonian $\mathcal{H}^{\text{eff}} + \mathcal{H}^{\text{loc}}$ with $J_1$ only obtained by the simulated

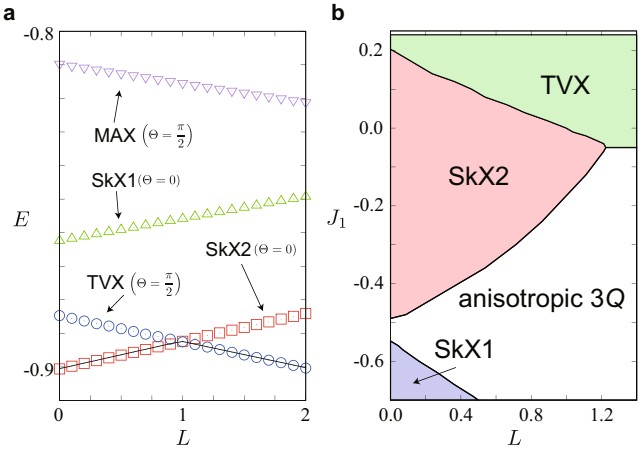

**Fig. 4 Ground states of the effective spin model. a** $L$ dependence of the variational energy per site, $E$, at $J = 1$ and $K = 0.4$. The solid line shows the result by the simulated annealing down to temperature $T = 10^{-4}$ for $N = 96^2$. **b** Phase diagram on the $L$–$J_1$ plane obtained by the simulated annealing.

annealing. While increasing $J_1$, the phase boundary between the SkX2 and TVX is shifted to a smaller $L$ rapidly. We note that the SkX2 turns into the SkX1 while decreasing $J_1$, whose tendency is similar to the results in Fig. 2a, b. In addition, we obtain the phase transition from the SkX1 to the anisotropic $3Q$ state while increasing $L$, similar to the result in Fig. 3b, which also indicates that $L$ plays a similar role to the temperature.

## Discussion

We have theoretically guided a new direction of exploring further exotic magnetic states beyond the SkXs by using the phase degree of freedom among the constituent spin density waves. The phase shift turns the SkXs into other states represented by the TVX, which are characterized by staggered emergent magnetoelectric fields and breaking of the lattice rotational symmetry. We unveiled three microscopic mechanisms which drive such a phase shift in itinerant magnets: the exchange interactions between the localized spins, thermal fluctuations, and the long-range chirality interactions.

Our results indicate that the skyrmion-based physics arising from nonzero net chirality can be switched on and off by changing the relative phases among the constituent waves. Furthermore, the lowering of the rotational symmetry by the phase shift induces nonreciprocal transport even in centrosymmetric systems without the spin-orbit coupling. These features open a new direction of emergent electromagnetism by the phase shift engineering. This would be realized in centrosymmetric skyrmion-hosting materials where the multiple-spin interactions rooted in the spin-charge coupling might play an important role[11,38,39]. While it is not straightforward to identify the phase shift by diffraction techniques such as the neutron scattering and the resonant x-ray scattering, our findings suggest that the angle-resolved photoemission spectroscopy and transport measurements will give good probes to detect the phase shift and new phases like the TVX.

Furthermore, the concept of the phase shift is not limited to the field of skyrmionics but ubiquitously useful for a variety of topological spin crystals, such as vortex crystals and hedgehog crystals, since they are characterized by the multiple-$Q$ spin density waves. Our results suggest that the overlooked phase degree of freedom can induce further interesting topological phase transitions, unconventional electronic structures, topological properties, and conductive phenomena, which will stimulate future exploration of functional spintronics materials in both experiment and theory.

## Methods

**Kondo lattice model**. We consider the Kondo lattice model on a triangular lattice, whose Hamiltonian is given by

$$\mathcal{H} = - \sum_{i,j,\sigma} t_{ij} c_{i\sigma}^{\dagger} c_{j\sigma} + J_{\mathrm{K}} \sum_{i} \mathbf{s}_i \cdot \mathbf{S}_i - H \sum_i S_i^z. \quad (1)$$

The first term represents the kinetic energy of itinerant electrons, where $c_{i\sigma}^{\dagger}$ ($c_{i\sigma}$) is a creation (annihilation) operator of an itinerant electron at site $i$ and spin $\sigma$. The second term represents the exchange coupling between itinerant electron spins $\mathbf{s}_i = (1/2)\sum_{\sigma,\sigma'} c_{i\sigma}^{\dagger} \boldsymbol{\sigma}_{\sigma\sigma'} c_{i\sigma'}$ [$\boldsymbol{\sigma} = (\sigma^x, \sigma^y, \sigma^z)$ is the vector of Pauli matrices] and classical localized spins $\mathbf{S}_i$ with $|\mathbf{S}_i| = 1$. The third term represents the Zeeman coupling to an external magnetic field $H$. In the calculations, we take the model parameters common to those in ref. [14]: the nearest- and third-neighbor hoppings, $t_1 = 1$ and $t_3 = -0.85$, respectively, $J_{\mathrm{K}} = 1$, and the chemical potential $\mu = -3.5$, which gives the characteristic wave vectors at $\mathbf{Q}_{\nu}$ ($\nu = 1, 2, 3$) with $Q = \pi/3$ in the main text (we take the lattice constant unity). In this parameter set, the ground state at zero field becomes the SkX2 in Fig. 1e, and that in a field becomes the SkX1 in Fig. 1c[14]. Note that the SkX2 is stable for other values of $\mathbf{Q}_{\nu}$ while changing the hopping parameters and the electron filling[14]; our arguments on the phase shift in the main text are not limited to $Q = \pi/3$ but can be applied to such other cases. To identify the spin and chirality structure, we compute the spin structure factor

$$S_s(\mathbf{q}) = \frac{1}{N} \sum_{\alpha = x,y,z} \sum_{j,l} S_j^{\alpha} S_l^{\alpha} e^{i\mathbf{q} \cdot (\mathbf{r}_j - \mathbf{r}_l)}, \quad (2)$$

and the chirality structure factor

$$S_{\chi}(\mathbf{q}) = \frac{1}{N} \sum_{\mu} \sum_{\mathbf{R}, \mathbf{R}' \in \mu} \chi_{\mathbf{R}} \chi_{\mathbf{R}'} e^{i\mathbf{q} \cdot (\mathbf{R} - \mathbf{R}')}, \quad (3)$$

respectively, where $\mu = (u, d)$ represent upward and downward triangles, respectively.

**Variational calculation for the Kondo lattice model**. In the variational calculations in Fig. 2a, b, we compare the energy of the following spin textures as the variational states: the triple spiral and sinusoidal crystals in Fig. 1c–f while varying three $\theta_{\nu}$ under the constraint $|\mathbf{S}_i| = 1$ at each site, the single-$Q$ spiral state characterized by $\mathbf{S}_i = (\cos \mathbf{q} \cdot \mathbf{r}_i, \sin \mathbf{q} \cdot \mathbf{r}_i, 0)$ where $\mathbf{q} = (0, 0), (Q_1, 0), (4\pi/3, 0)$ denoted as $1Q$ K and $(0, 2\pi/\sqrt{3})$ denoted as $1Q$ M, and the conical ($1Q$ C) state characterized by $\mathbf{S}_i = (1/N_m)(\cos \mathbf{Q}_1 \cdot \mathbf{r}_i, \sin \mathbf{Q}_1 \cdot \mathbf{r}_i, a_z)$ where $N_m$ is the normalization factor and $a_z$ is the variational parameter. We assume $Q = \pi/3$ in the variational calculations, since it was shown that the spin states with $Q = \pi/3$ give the lowest grand potential by performing the unbiased Langevin dynamics simulations with the kernel polynomial method when the exchange interactions between the localized spins are zero[14]. The phase diagram is obtained for the system size $N = 96^2$ under the periodic boundary condition.

**Finite-temperature calculation for the Kondo lattice model**. We adopt the Langevin dynamics simulation with the kernel polynomial method[27] to study the finite-temperature properties of the Kondo lattice model in Fig. 3. In the kernel polynomial method, we expand the density of states by up to 2000th order of the Chebyshev polynomials with $16^2$ random vectors. In the Langevin dynamics, we use a projected Heun scheme for 1000−5000 steps with the time interval $\Delta \tau = 2$. The simulations are done for $N = 60^2$, $72^2$, $96^2$, and $120^2$ sites, and the thermal averages are taken for 100−800 samplings after the thermalization. In the main text, we show the results for $N = 96^2$ and $120^2$, as those for $N \geq 72^2$ are convergent within the error bars. The data at different temperatures are obtained independently starting from different random states. In the simulations, we compute the **q** components of the scalar chirality $\chi_{\mathbf{q}} = \sqrt{S_{\chi}(\mathbf{q})/N}$ at $\mathbf{q} = 0$, $2\mathbf{Q}_1$, $2\mathbf{Q}_2$, and $2\mathbf{Q}_3$, as plotted in Fig. 3a.

**Effective spin model**. An effective spin model, which is derived from the Kondo lattice model in equation (1), is given by[35]

$$\mathcal{H}^{\mathrm{eff}} = 2 \sum_{\nu=1}^{3} \left[ -J\mathbf{S}_{\mathbf{Q}_\nu} \cdot \mathbf{S}_{-\mathbf{Q}_\nu} + \tilde{K}(\mathbf{S}_{\mathbf{Q}_\nu} \cdot \mathbf{S}_{-\mathbf{Q}_\nu})^2 \right] + \tilde{L} \left[ \left\{ \mathbf{S}_{\mathbf{Q}_1} \cdot (\mathbf{S}_{\mathbf{Q}_2} \times \mathbf{S}_{\mathbf{Q}_3}) \right\}^2 + \mathrm{H.c.} \right], \quad (4)$$

where $\mathbf{S}_{\mathbf{Q}_\nu} = (1/\sqrt{N}) \sum_i \mathbf{S}_i e^{i\mathbf{Q}_\nu \cdot \mathbf{r}_i}$. The first two terms describe bilinear and biquadratic interactions, which are derived by second- and fourth-order perturbation expansions in terms of the spin-charge coupling, respectively[35]; $J > 0$ and $\tilde{K} = K/N > 0$, and $N$ denotes the number of sites. The $J$ and $K$ terms provide a minimal effective model for the Kondo lattice model, stabilizing the SkX2 at zero field (Fig. 3c) and the SkX1 at finite fields (Fig. 3e)[35]. Meanwhile, the third term with $\tilde{L} = L/N^2$ represents an interaction between the scalar spin chirality composed of $\mathbf{S}_{\mathbf{Q}_\nu}$ (see Supplementary Information).

**Variational calculation and simulated annealing for the effective spin model**. In the variational calculations in Fig. 4a, we compare the energy of the four states in Fig. 1c–f while varying three $\theta_{\nu}$ under the constraint $|\mathbf{S}_i| = 1$ at each site. The results are for $J = 1$ and $K = 0.4$ while changing $L$ for the system with $N = 96^2$ sites. In the simulated annealing in Fig. 4a, b, our simulations are carried out with the standard Metropolis local updates in real space, by reducing the temperature successively, from

$T = 0.1 - 1.0$ to $\simeq 10^{-4}$ with the cooling rate of $0.99995 - 0.99999$. The final temperature is typically $T = 10^{-4}$. Each phase is identified by its spin and chirality configurations. We also start the simulations from the spin structures obtained at low temperatures to determine the phase boundaries between different magnetic states.

## Data availability

The data that support the findings of this study are available from the corresponding author upon reasonable request.

## Code availability

The codes used for this study are available from the corresponding author upon reasonable request.

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

## Acknowledgements

The authors thank R. Ozawa for the fruitful discussions. This research was supported by JSPS KAKENHI Grants Numbers JP18H04296 (J-Physics), JP18K13488, JP19K03752, JP19H01834, 19K03740, JP19H05822, JP19H05825, and JP21H01037, JST CREST (JP-MJCR18T2), and JST PRESTO (JP-MJPR1912 and JP-MJPR20L8). This work was also supported by the Toyota Riken Scholarship. T.O. acknowledges support by the Endowed Project for Quantum Software Research and Education, The University of Tokyo (https://qsw. phys.s.u-tokyo.ac.jp/). Parts of the numerical calculations were performed in the supercomputing systems in ISSP, the University of Tokyo.

## Author contributions

S.H. and Y.M. conceived the project. S.H. performed the numerical and analytical calculations. All the authors interpreted the results and wrote the manuscript.

## Competing interests

The authors declare no competing interests.
