## [Peer Review File · Nature Communications]

Reviewers' Comments:

Reviewer #1:

None

Reviewer #2:

Remarks to the Author:

The manuscript of Hayami et al entitled "Phase shift in skyrmion crystals" is a nice theoretical study of various skyrmion/spin-swirling lattices based on the view that they result from the interference of spin density waves. By defining a phase shift between the waves building-blocks, the investigation explores how various lattices of complex spin-textures are related to each other as well as their basic properties encoded in the three-spin scalar chirality. For instance, how one can "convert" a skyrmion lattice to a moron-antimeron crystal by modifying the phase shift, which as discussed by the authors, is controlled by the magnitude of Heisenberg interactions or tuned by thermal fluctuations. The stability analysis is based on an extended Heisenberg model with the use of a Kondo lattice model with a well defined set of phenomenological parameters chosen by the authors. I find the paper well written and the paper of interest for experts in the field of skyrmionics but I do not find it suitable for Nature Communications. The idea of using the phase shifts between the constituent spin density waves to discuss the relation between the different complex spin textures lattices is certainly interesting but I find that the presented results more suitable for a more specialised journal.

Reviewer #3:

Remarks to the Author:

This paper presents numerical studies on a Kondo lattice model on the triangular lattice, with the goal of identifying the conditions for stabilizing new 3Q states related to skyrmion crystals. These 3Q states are obtained from the conventional spiral or sinusoidal 3Q states by shifting the phase of each component by the same amount. Both the subject of the study and the new phases identified in this work are interesting. The work may be published in Nature Communications if the authors could address the following issues:

(1) The constraint that the three phase shifts are equal seems to be artificial. Since a global coordinate shift by an arbitrary Bravais lattice vector should not be a relevant degree of freedom, only two of the three phase shifts are independent. When performing variational calculations, if these two phase shifts are treated as independent variational parameters, the phase diagrams may be very different. The authors should provide justifications on why they enforce this constraint.

(2) The phases based on the 3Q sinusoidal state discussed in this work seem to be very specialized, since they require special values of Q in order to satisfy $|S_i|=1$. For these Q , only a few values of the phase shift can satisfy the constraint $|S_i|=1$. It is therefore questionable whether one should treat the phase shift as a continuous variational parameter once the constant length constraint is enforced.

(3) The value of $Q=\pi/3$ is only mentioned in Methods. It seems that the authors just considered a special Kondo lattice model that favors such an ordering wave vector magnitude, because of which they did not take the size of Q as a variational parameter in their calculations. If this is indeed the case, the authors should make this limitation explicit and discuss how their conclusions are relevant in more general cases, especially for the spin-spiral based 3Q states whose Q can be arbitrary.

=== Response to Reviewer #2

Reviewer #2 (Remarks to the Author):

The manuscript of Hayami et al entitled "Phase shift in skyrmion crystals" is a nice theoretical study of various skyrmion/spin-swirling lattices based on the view that they result from the interference of spin density waves. By defining a phase shift between the waves building-blocks, the investigation explores how various lattices of complex spin-textures are related to each other as well as their basic properties encoded in the three-spin scalar chirality. For instance, how one can "convert" a skyrmion lattice to a moron-antimeron crystal by modifying the phase shift, which as discussed by the authors, is controlled by the magnitude of Heisenberg interactions or tuned by thermal fluctuations. The stability analysis is based on an extended Heisenberg model with the use of a Kondo lattice model with a well defined set of phenomenological parameters chosen by the authors. I find the paper well written and the paper of interest for experts in the field of skyrmionics but I do not find it suitable for Nature Communications. The idea of using the phase shifts between the constituent spin density waves to discuss the relation between the different complex spin textures lattices is certainly interesting but I find that the presented results more suitable for a more specialised journal.

We thank Reviewer #2 for reviewing our manuscript and recognising the novelty of our work by saying "The idea of using the phase shifts between the constituent spin density waves to discuss the relation between the different complex spin textures lattices is certainly interesting". On the other hand, he/she recommended the publication in a more specialised journal. We would like to draw the reviewer's attention to the fact that the concept of the phase shift unveiled in our study is not limited to the field of skyrmionics but ubiquitously useful for a variety of topological spin crystals, such as vortex crystals and hedgehog crystals. This will stimulate future experimental and theoretical studies of unconventional electronic structures, topological properties, and conductive phenomena through the overlooked phase degree of freedom. We would also like to stress that this work provides the first theoretical framework for the phase shift in such topological spin crystals that leads to new types of phase transitions. We have revised the manuscript to clearly state these points. [See the summary of changes (1).]

We strongly believe that our work will make a profound impact on the wide range of active research fields for magnetism, topological physics, and spintronics. We hope that the reviewer recognises the general interest of our study and reconsiders the publication in Nature Communications.

=== Response to Reviewer #3

Reviewer #3 (Remarks to the Author):

This paper presents numerical studies on a Kondo lattice model on the triangular lattice, with the goal of identifying the conditions for stabilizing new 3Q states related to skyrmion crystals. These 3Q states are obtained from the conventional spiral or sinusoidal 3Q states by shifting the phase of each component by the same amount. Both the subject of the study and the new phases identified in this work are interesting. The work may be published in Nature Communications if the authors could address the following issues:

We thank Reviewer #3 for reviewing our manuscript and for recognising the importance of our study. We also appreciate his/her helpful comments to improve our paper. We have revised the manuscript following the reviewer's suggestions. Below are the responses to the reviewer's comments.

(1) The constraint that the three phase shifts are equal seems to be artificial. Since a global coordinate shift by an arbitrary Bravais lattice vector should not be a relevant degree of freedom, only two of the three phase shifts are independent. When performing variational calculations, if these two phase shifts are treated as independent variational parameters, the phase diagrams may be very different. The authors should provide justifications on why they enforce this constraint.

As the reviewer pointed out, only two of the three phases are independent, each of which modifies the energy of the skyrmion crystals. Actually, we have already taken into account the individual phase degrees of freedom and have chosen the optimal phases in the variational calculations, although we did not explicitly described them in the previous manuscript. We have added a figure for this consideration in Supplementary Information. The data clearly show that the grand potential smoothly varies while changing the two phases, and there are minima at a particular set of the phases, which were adopted in the variational calculations in the main text. Hence, our conclusions remain valid even when the phases are fully optimised. [See the summary of changes (5).]

(2) The phases based on the 3Q sinusoidal state discussed in this work seem to be very specialized, since they require special values of Q in order to satisfy $|\mathbf{S}_i|=1$. For these Q , only a few values of the phase shift can satisfy the constraint $|\mathbf{S}_i|=1$. It is therefore questionable whether one should treat the phase shift as a continuous variational parameter once the constant length constraint is enforced.

In general, the constraint of $|\mathbf{S}_i|=1$ is not satisfied for a superposition of more than a single spin density wave. This also holds for the state with $Q=\pi/3$ discussed in our study. Hence, in both sinusoidal and

spiral cases, we normalise the spin length at each site in the variational state with given Q and phases. In this framework, one can prepare the variational state with arbitrary phases. In other words, one can treat the phases as continuous variables in the variational calculations. A particular set of the phases will be selected by the optimization of the energy, as shown in the manuscript [see also the reply to (1)]. The situation is common to the sinusoidal and spiral cases. We have added sentences about the above argument in Supplementary Information. [See the summary of changes (4).]

(3) The value of $Q=\pi/3$ is only mentioned in Methods. It seems that the authors just considered a special Kondo lattice model that favors such an ordering wave vector magnitude, because of which they did not take the size of Q as a variational parameter in their calculations. If this is indeed the case, the authors should make this limitation explicit and discuss how their conclusions are relevant in more general cases, especially for the spin-spiral based $3Q$ states whose Q can be arbitrary.

As the referee pointed out, we assume $Q=\pi/3$ in the variational calculations in Fig. 2. This is because we obtained the spin states with $Q=\pi/3$ by performing the unbiased Langevin dynamics simulations with the kernel polynomial method when the exchange interactions between the localised spins are zero. From the result, we postulate that the spin states with other q have a higher energy than that with $Q=\pi/3$. Meanwhile, we take into account the spin states with $q=(4\pi/3,0)$ and $q=(0,2\pi/\sqrt{3})$, since the interactions tend to favor these states. We have added a sentence to describe the details of the variational calculations in Methods. [See the summary of changes (3).]

Besides, we would like to draw the reviewer's attention to the previous study that demonstrates the $3Q$ sinusoidal states are ubiquitously found at different values of Q . See Supplemental Material in Ref. [14]. There, the $3Q$ states are stabilised at $Q=0.18\pi$ and 0.25π , which are different from $Q=\pi/3\sim 0.33\pi$ (in the present paper), by taking different electron fillings. This is because the $3Q$ sinusoidal spin states appear irrespective of the value of Q when the bare susceptibility shows global maxima at the corresponding Q . In this respect, we expect that a similar phase shift in the skyrmion crystals can take place in a wide parameter region. We have added a sentence on this issue in Methods. [See the summary of changes (2).]

We hope that the revisions we made meet the reviewer's request.

====Summary of changes

- (1) In page 11 of the main text (2nd paragraph): We added a paragraph to stress that our results are of interest from a broad perspective.
- (2) In page 12 of the main text (Methods of "Kondo lattice model"): We added a sentence to state the applicability of our arguments to the SkX_2 with other characteristic wave vectors.
- (3) In page 13 of the main text (Methods of "Variational calculation for the Kondo lattice model"): We added a sentence to describe the details of the variational calculations.
- (4) In page 1 of Supplementary Information: We have added sentences to describe the details of the phases in the variational states for the $3Q$ sinusoidal and spiral cases.
- (5) In page 2 of Supplementary Information: We added Fig. S2 to represent the optimal phases in the $3Q$ sinusoidal and spiral states. Accordingly, we modified sentences in page 1.

Reviewers' Comments:

Reviewer #2:

Remarks to the Author:

The authors addressed my worries concerning the manuscript being not suitable for a journal like Nature Communications. Considering their reply to the comments and questions of both referees, I recommend now its publication.

Reviewer #3:

Remarks to the Author:

The authors have clarified the points I mentioned in my previous report. I would like to recommend the publication of the present version of the manuscript in Nature Communications.

To Reviewer #2

The authors addressed my worries concerning the manuscript being not suitable for a journal like Nature Communications. Considering their reply to the comments and questions of both referees, I recommend now it publication.

[Response]

We are pleased that the reviewer evaluated our manuscript by stating “I recommend now it publication.”

To Reviewer #3

The authors have clarified the points I mentioned in my previous report. I would like to recommend the publication of the present version of the manuscript in Nature Communications.

[Response]

We are pleased that the reviewer evaluated our manuscript by stating “*I would like to recommend the publication of the present version of the manuscript in Nature Communications.*”